# ASR Language Resources for Faroese

**Carlos Mena**
Language and Voice Lab,
Reykjavík University,
102 Reykjavík, Iceland
carlosm@ru.is

**Annika Simonsen**
University of Iceland,
107 Reykjavík, Iceland
annika@hi.is

**Jón Guðnason**
Language and Voice Lab,
Reykjavík University,
102 Reykjavík, Iceland
jg@ru.is

## Abstract

The aim of this work is to present a set of novel language resources in Faroese suitable for the field of Automatic Speech Recognition including: an ASR corpus comprised of 109 hours of transcribed speech data, acoustic models in systems such as WAV2VEC2, NVIDIA-NeMo, Kaldi and PocketSphinx; a set of n-gram language models and a set of pronunciation dictionaries with two different variants of Faroese. We also show comparison results between the distinct acoustic models presented here. All the resources exposed in this document are publicly available under creative commons licences.

## 1 Introduction

As the digital world has become increasingly prominent and omnipresent in most human activities, the use of more and better language technologies has become a pressing need. For this reason, more and more governments are investing in the development of all kinds of linguistic resources that allow their citizens to be part of the new digital era, with all the benefits it entails. Language technology initiatives in the main regions of the world such as: Europe (Rehm et al., 2020; Nikulásdóttir et al., 2020; Meister et al., 2010; D'Halleweyn et al., 2006), India (Vikas, 2001; Choudhary, 2021), Africa (Grover et al., 2011), China (Kania et al., 2018), Saudi Arabia (Maegaard et al., 2008, 2005) and the Spanish speaking countries (Fernandez et al., 2016); allow us to attest how important language technologies have become in recent times.

In synchrony with all the developments mentioned above, it is time to talk about the efforts made for the development of the Faroese language in the digital sphere. The most recent initiative in this regard is the Ravnur Project, founded in the Faroe Islands. Thanks to the resources generated and shared by Ravnur, it has been possible to develop all the language resources presented in this document.

### 1.1 Faroese

The Faroe Islands is a set of small islands located at the North Atlantic in a half way between Scotland, Iceland and Norway. It is an autonomous territory of the Kingdom of Denmark with Faroese as the official language, which is spoken by around $54,000$ people. There are four main dialect areas in the Faroe Islands; north, northwest, central and southern (Petersen, 2022). The Faroe Islands is a bilingual country with Danish as the second official language. While many native speakers of Faroese use Danish for university education or employment in Denmark, Faroese is spoken as a first language by most of the population and is used on all domains, e.g. in education, public sectors, church etc. in the Faroe Islands. The first and, to this date, only Faroese speech synthesis was created in 2005 (Helgason and Gullbein, 2005) by combining efforts from researchers at the University of Stockholm and the University of the Faroe Islands and is used by the visually impaired community. Currently, there is a huge demand for Faroese ASR solutions, needed by the deaf, visually impaired and dyslexic communities - and also the general public, who wish to use their mother tongue when interacting with technology.

### 1.2 The Ravnur Project

The Faroese ASR research project, *Ravnur*, was assembled in 2019 (Foundation, 2019). The aim of the project was to create open-source resources that could be used to create automatic speech recognition (ASR) systems in Faroese. These resources would also be useful for creating other types of language technologies, as well as for lin-

guistic research. The project was funded by public and private initiators and investors, including the Faroese government. The development team consisted of a project leader, a technical leader, three native speaking junior linguists, an IT assistant, five university student assistants, as well as external advisors. The project concluded in the summer of 2022 with the publication of the Basic Language Resource Kit for Faroese (BLARK) (Simonsen et al., 2022; Debess et al., 2022).

## 1.3 Collection of the Speech Corpus

A Basic Language Resource Kit or BLARK is defined as the minimal set of language resources needed to create language and speech technology for a language (Krauwer, 2003; Maegaard et al., 2006). A BLARK is ideally language independent, but because languages may have different requirements, the contents of the BLARK may vary in some respects from language to language.

So, as Ravnur was an ASR project, the focus was on collecting good quality recordings of Faroese and creating a transcription corpus and pronunciation dictionary. During the course of the project, Ravnur collected 135 hours of recordings of 433 speakers total (249 female speakers and 184 male speakers) reading text of various genres, such as news, blogs, Wikipedia, law texts, GPS commands, word lists etc. The participants self-reported their gender, native language, dialect and age which varies between 15 to 83 years old. The recordings were made on TASCAM DR-40 Linear PCM audio recorders using the built-in stereo microphones in WAVE 16 bit with a sample rate of 48kHz. All recordings have been manually orthographically transcribed, while part of the speech corpus has been phonetically transcribed. The transcriptions were made by the university student assistants and the three Faroese linguists working for the project. All words that occur in the recordings were put in a pronunciation dictionary. The dictionary includes phonetic transcriptions written in SAMPA and PAROLE PoS-tags (Bilgram and Keson, 1998; Keson, 1998)[1].

As it can be seen, the BLARK developed by Ravnur is the starting point of the novel machine learning models presented in this work.

## 2 The Ravnursson Corpus

Ravnursson[2] (Hernández Mena and Simonsen, 2022) is an ASR corpus with a length of 109 hours[3], extracted from the BLARK described in section 1.3. Unlike the original BLARK, the Ravnursson only contains the speech files along with their respective transcriptions. The main characteristics of the corpus are the following:

- The audio files in this corpus are distributed in a FLAC format at 16kHz@16bit mono.

- The corpus contains 71,949 speech files from 433 speakers.

- The corpus is split into train, dev, and test portions. Lengths of every portion are: train = 100h08m, dev = 4h30m, test = 4h30m.

- The development and test portions have exactly 10 male and 10 female speakers each and both portions have exactly the same size in hours.

- As the test and development portions were selected to be gender balanced, an equal representation of all the dialectal variants is not guarantee in these two portions.

- Due to the limited number of prompts to read, only 39,945 of the 71,949 prompts in the whole corpus are unique. In other words, 44.48% of the prompts in the corpus are repeated at least once.

- Despite the repeated prompts in the corpus, the development and test portions do not share speakers with each other or with the training set.

### 2.1 Analysis of the Repeated Prompts

As the number of reading prompts for the corpus was limited during the recording process, the common denominator in the Ravnursson corpus is that one prompt is read by more than one speaker. This is relevant because it is a common practice in ASR

---

[1]Both the Faroese SAMPA alphabet (sometimes called FARSAMPA) and PAROLE PoS-tags were created by Ravnur for the BLARK.

[2]As a matter of fact, the name Ravnursson comes from Ravnur (a tribute to the Ravnur Project) and the suffix "son" which in Icelandic means "son of". Therefore, the name "Ravnursson" means "The (Icelandic) son of Ravnur". The double "ss" is just for aesthetics.

[3]As it was mentioned in section 1.3, 135 hours of speech data were collected for the original BLARK. However, the Ravnursson Corpus contains 109 hours because we removed the portions with no presence of speech as much as we could.

to create a language model using the prompts that are found in the train portion of the corpus. That is not recommended for the Ravnursson Corpus as it counts with several prompts shared by all the portions and that will produce an important bias in the language modeling task.

Table 1 shows some statistics about the repeated prompts through all the portions of the corpus. The way this table has to be understood is as follows: for example, the first row indicates that there is a total of $71,949$ reading prompts in the whole corpus; $39,945$ of those are unique and $32,004$ are repeated at least once. Therefore, a total of $44.48\%$ prompts in the whole corpus are repeated at least once. The same applies to the rest of the rows in Table 1.

| Corpus Portion | Total Prompts | Unique Prompts | Repeat. Prompts | % |
|---|---|---|---|---|
| **All** | $71,949$ | $39,945$ | $32,004$ | $44.48\%$ |
| **Train** | $65,616$ | $38,646$ | $26,970$ | $41.1\%$ |
| **Test** | $3,002$ | $2,887$ | $115$ | $3.83\%$ |
| **Dev** | $3,331$ | $3,302$ | $29$ | $0.87\%$ |

Table 1: Analysis of Repeated Prompts.

## 2.2 Corpus Organization

The "speech" directory contains all the speech files of the corpus. The files in the speech folder are divided in three directories: train, dev and test. The train portion is sub-divided in three types of recordings: RDATA1O, RDATA1OP and RDATA2; this is due the organization of the recordings in the original BLARK. There, the recordings are divided in Rdata1 and Rdata2.

One main difference between Rdata1 and Rdata2 is that the reading environment for Rdata2 was controlled by a software called "PushPrompt" which is included in the original BLARK (Simonsen et al., 2022). Another difference is that in Rdata1 there are some available transcriptions labelled at the phoneme level. The audio files in the speech directory of the Ravnursson corpus are divided in the folders RDATA1O where "O" is for "Orthographic" and RDATA1OP where "O" is for Orthographic and "P" is for phonetic. These categories are just a reminiscence of the original BLARK but it does not imply that the Ravnursson corpus comes with transcriptions at the phonetic level. In the case of the dev and test portions, the data come only from Rdata2 which does not have

labels at the phonetic level in the original BLARK.

## 2.3 The Metadata File

The metadata file is a "tab-separated values file" (TSV) containing all the relevant information of the corpus. The file can be read using the Pandas (McKinney et al., 2010) library in Python and it comprises of the following 12 columns:

1. id: The filename without the extension ".flac".

2. speaker_id: The filename without the segment number.

3. filename: Full filename including the extension ".flac".

4. sentence_norm: The normalized transcription: no punctuation marks, no digits, lower case letters, one single space between words.

5. gender: The gender of the speaker: male or female.

6. age: The age range of the speaker: 15-35, 36-60, 61+ years old.

7. native_language: "Faroese" in all the cases.

8. dialect: The speaker dialect.

9. created_at: The date when the audio file was recorded.

10. duration: Duration of the speech file in seconds.

11. sample_rate: $16kHz$ in all the cases.

12. status: The corpus portion: train, test or dev.

## 2.4 Codification of the Audio Filenames

In the Ravnursson corpus, the filenames of the audio files encode relevant information about the respective speech files. The first row of Table 2, shows a typical audio filename. The second row enumerates the fields of information encoded in the filename and the third row shows the same filename of row one but broken down in the eight parts as specified in the second row.

The explanation of the information encoded in the filename is at follows:

1. Gender of the Speaker: **M** for male or **K** for female

| MEY01_040319_rok0_0009.flac | | | | | | | |
|---|---|---|---|---|---|---|---|
| **1** | **2** | **3** | **4** | **5** | **6** | **7** | **8** |
| M | E | Y | 01 | 040319 | rok0 | 0009 | .flac |

Table 2: Audio Filename Format.

2. Dialect Group: **U** for Suðuroy, **A** for Sandoy, **S** for Suðurstreymoy, **E** for Norðurstreymoy/Eysturoy (exclusive of Eiði, Gjógv og Funningur), **V** for Vágar and **N** for Norðuroyggjar (inclusive of Eiði, Gjógv og Funningur)

3. Age Group: **Y** for "Younger" between 15-35 years old, **M** for "Middle-aged" between 36-60 years old and **E** for "Elderly" 61 years old or older.

4. Number of Speaker in a Group: is a number that always consists of two digits and starts with 01, 02, 03 etc. The first speaker in a group with the same gender, dialect group and age group (e.g. MEY) gets the number 01. The next speaker in the same group gets the number 02 (and his ID is therefore MEY02).

5. Date: The date when the speech was recorded (day/month/year).

6. Type of reading material: This code can only be found in speech files at RDATA1O and RDATA1OP. For more information about the types of reading material please see the documentation of the original BLARK and its directory "readingtexts_1.0".

7. Segment Number: In the original BLARK the recording session is distributed as one audio file per speaker and it can be very long from the ASR perspective. So, the audio files are subdivided in segments of around 10 seconds to fit most of the modern ASR engines[4]. The numbering is continuous for each speaker; the only exception is with the files MUY01_180519_set4_0004 and MUY02_190120_eind2_0007. We de-

tected that they are empty and we removed them.

8. File extension: The corpus is distributed in FLAC format.

# 3 Acoustic Models

The development of the Ravnursson corpus allowed us to create acoustic models in four different ASR systems: WAV2VEC2, NeMo, Kaldi and PocketSphinx. In this section we discuss the details of how we created each of them.

## 3.1 WAV2VEC2 Model

WAV2VEC, released in 2019, is a convolutional neural network that takes raw audio as input and computes a general representation that can be input to a speech recognition system (Schneider et al., 2019). In 2020, a second version, WAV2VEC2 (Baevski et al., 2020) was released. Based on WAV2VEC2, the XLSR-53 (Conneau et al., 2020) was also released in 2020. XLSR-53 is a open-source model trained with more than 50k hours of unlabelled speech in 53 languages. It can be used to create acoustic models in any language through a fine-tuning step.

Using the XLSR-53 as a starting point, we created an acoustic model suitable for Faroese (Hernandez Mena, 2022b)[5] which is available on a Creative Commons licence "CC BY 4.0". The fine-tuning process for this model lasted 30 epochs. Due to the acceptable WER results that we obtained with this model, we decided not to add any type of augmentation to the training data.

## 3.2 NeMo Model

NeMo (Neural Modules) is a Python toolkit developed by NVIDIA for creating AI applications. It comes with extendable collections of pre-built modules for automatic speech recognition and natural language processing (Kuchaiev et al., 2019). One of the NeMo modules suitable for speech recognition is called Quartznet (Kriman et al., 2020) which is a convolutional model trained with Connectionist Temporal Classification (Graves, 2012) or CTC for short.

In order to train an ASR model for Faroese in NeMo, we used the public checkpoint

---

[4]According to the developers of Sphinx, the optimal length for audio recordings in ASR is between 5 and 30 seconds (see https://cmusphinx.github.io/wiki/tutorialam/. However, we segmented the audio files of the Ravnursson Corpus to have a lenght around 10 seconds to fit the format of other corpora developed by our laboratory

---

[5]Available at: https://huggingface.co/carlosdanielhernandezmena/wav2vec2-large-xlsr-53-faroese-100h

|  | Points of articulation | | | | | | | | |
|---|---|---|---|---|---|---|---|---|---|
| **Consonants** | **Bi-labial** | **Labiodental** | **Dental** | **Alveolar** | **Post-alveolar** | **Retroflex** | **Palatal** | **Velar** | **Glottal** |
| Voiceless Stop | p | | | t | | | | k | |
| Voiced Stop | b | | | d | | | | g | |
| Voiceless Affricate | | | | | tS | | | | |
| Voiced Affricate | | | | | dZ | | | | |
| Voiceless Fricative | | f | 5 | s | S | z | | | h |
| Voiced Fricative | | v | 4 | | | | | | |
| Voiceless Nasal | M | | | x | | | | X | |
| Voiced Nasal | m | | | n | | | | N | |
| Voiceless Lateral | | | | L | | | | | |
| Voiced Lateral | | | | l | | | | | |
| Approximants | | | | r | | | j | w | |
| | | | | | | | | | |
| **Vowels** | | | | | **Front** | | **Central** | | **Back** |
| Close | | | | | i y | | 3 | | u |
| | | | | | | I Y | | U | |
| Close-mid | | | | | e | 2 | | | o |
| | | | | | | | 8 | | |
| Open-mid | | | | | | E 9 | | | O |
| | | | | | | | | | |
| Open | | | | | | a | | | |

Table 3: Phonetic Repertoire of Faroese

"QuartzNet15x5Base-En.nemo[6]" as a starting point. This model was trained with more than 3k hours of English data in a Quartznet architecture during 600 epochs. Based on a work by Huang et al.[7], we fine-tuned the checkpoint with the data of the Ravnursson corpus during 236 epochs, obtaining a first checkpoint able to recognize Faroese. Then, we augmented the initial 100 hours of the training portion of the Ravnursson corpus to 300 hours through speech perturbation using two speed rates: 0.9 and 1.1. Finally, we fine-tuned our initial checkpoint in Faroese with the augmented data during 163 epochs to obtain a final model[8] (Hernandez Mena, 2022a) which is available on a Creative Commons licence "CC BY 4.0".

### 3.3 Kaldi Model

Kaldi (Povey et al., 2011), released in 2011, is a well established toolkit for speech recognition written in C++, which is based on distinct paradigms such as: finite-state transducers (Allauzen et al., 2007), Hidden Markov Models (Juang and Rabiner, 1991), Gaussian Mixture Models (Naeem et al., 2020) as well as neural networks (Rath et al., 2013).

Our "Kaldi Recipe for Faroese[9]" (Hernández Mena, 2022) was created using the Ravnursson corpus as training data. The recipe produces models based on Hidden Markov Models (HMMs) as well as Neural Networks; in specific, the neural network is an LSTM or "Long Short-Term Memory" (Huang et al., 2017) and it uses speed perturbation as augmentation technique with speed rates of 0.9 and 1.1. This recipe requires a 3-gram language model (lm) for decoding, a 4-gram lm for re-scoring and a pronouncing dictionary; elements that are available in our "Faroese Language Models with Pronunciations" (Hernández Mena et al., 2022), discussed in further sections.

---

[6]Available at: https://catalog.ngc.nvidia.com/orgs/nvidia/models/nemospeechmodels/files

[7]The decision of using the QuartzNet architecture and not others, was based mainly on this research paper. A comparison of different NeMo architectures is beyond the scope of this paper.

[8]Available at: https://huggingface.co/carlosdanielhernandezmena/stt_fo_quartznet15x5_sp_ep163_100h

[9]See: https://github.com/CarlosDanielMena/Kaldi_Recipe_for_Faroese

The recipe is available on `Clarin.is` [10] under a Creative Commons licence "CC BY 4.0".

## 3.4 PocketSphinx Model

Sphinx is an old speech recognition system based on Hidden Markov Models developed by Carnegie-Mellon University in the late 80's (Lee et al., 1990). Through time, progressive versions of Sphinx have been released up the version 4. At some point, the version 2 turned into PocketSphinx (Huggins-Daines et al., 2006). PocketSphinx was supposed to be a lighter and faster version of Sphinx but nowadays it has become the main version that can be used in real time mode, even in ARM processors. PocketSphinx has long ceased to be a suitable system for research, but nevertheless it still has an active community of users that choose it as a real time speech recognition system in devices with not a great computing power such as Raspberry PI (Upton and Halfacree, 2014) or other ARM computers.

Our PocketSphinx models[11], trained with the Ravnursson corpus, are suitable for the Pocket-Sphinx Python library available at the Pypi repository [12]. With this library it is possible to perform both standard and real time speech recognition, forced-alignment and produce timestamps. The version of PocketSphinx that was available when we produced these models was the number 4. Few weeks later the version 5 was released but our models remain compatible.

The example language model that comes with the PocketSphinx model is a 3-gram model created using the training prompts of the Ravnursson Corpus. The test portion of the corpus was used to measure a WER of 18.7%. We don't show this result in Table 5 because the use of the training prompts in the language model produces a bias that is not fair to the other models as we point out in section 2.1. We strongly recommend to create a language model for the specific task that is required and to kept it as short as possible because a larger model will impact the latency of the system.

## 4 Pronunciation Models

The pronunciation models that we discuss in this section is a set of pronouncing dictionaries that are included in our "Faroese Language Models with Pronunciations" (Hernández Mena et al., 2022) along with a number of language models that will be discussed in section 5. Most of the pronunciations come from the original BLARK, but for convenience, we subdivide them in different dictionaries as follows:

- Central_Faroese.dic: It contains pronunciations of the variant of Faroese which is spoken in the capital.

- East_Faroese.dic: It contains pronunciation of the northwest variant of Faroese[13].

- Ravnursson_Composite_Words.dic: It contains words with hyphens and/or underscores that are present in the Ravnursson Corpus. We keep them separate in a different dictionary because these type of composite words can be problematic for a grapheme-to-phoneme (g2p) tool.

- BLARK.dic: It contains pronunciations of words that are present in the BLARK but that are not present in any other dictionary of the set.

- FAROESE_ASR.dic: This dictionary is recommended for ASR experiments in Kaldi or any other ASR system based on phonemes. The dictionary is the mix of Central_Faroese.dic, East_Faroese.dic and Ravnursson_Composite_Words.dic. It is important to clarify that the dictionary can contain words with multiple pronunciations, which is normal in Kaldi-like systems.

### 4.1 Phoneme Sets of Dictionaries

Table 3 shows the phonetic repertoire of Faroese using 42 SAMPA symbols. Each of these correspond to an individual phoneme that is included

---

[10]See: `http://hdl.handle.net/20.500.12537/305`

[11]Available at: `https://github.com/CarlosDanielMena/RAVNURSSON_FAROESE_Models_100h`

[12]See: `https://pypi.org/project/pocketsphinx/`

[13]In the most recent dialect classification (Petersen, 2022), the islands in the northwest area are classified as being the same dialect area. However, there is a difference in the pronunciation of the digraph *ei* between the westernmost islands and the more central and eastern islands in that dialect area. Therefore, the westernmost part of the dialect area is not included in our EAST dictionary. For that reason, we have given this dictionary the name EAST. The idea is that this makes it is possible to make WEST, NORTH and SOUTH dictionaries in the future.

| SAMPA | IPA | SAMPA | IPA | SAMPA | IPA | SAMPA | IPA |
|--------|-----|--------|-----|--------|-----|--------|-----|
| p | pʰ | m | m | e | e | aJ | ai |
| b | b | M | m̥ | E | ɛ | aW | au |
| t | tʰ | n | n | a | a | OJ | ɔi |
| d | d | x | n̥ | y | y | OW | ɔu |
| k | kʰ | N | ŋ | Y | ʏ | 3W | ʉu |
| g | g | X | ŋ̊ | 2 | ø | EW | eu |
| f | f | l | l | 9 | œ | 9W | œu |
| v | v | L | l̥ | u | u | 9J | œi |
| s | s | j | j | o | o | 4 | ð |
| S | ʃ | w | w | O | ɔ | 5 | θ |
| z | ṣ | r | ɹ | EA | ɛa | 8 | ə |
| h | h | U | ʊ | OA | ɔa | H | Pre-aspiration |
| tS | tʃʰ | i | i | UJ | ʊi |  |  |
| dZ | ʥ | I | ɪ | EJ | ɛi |  |  |

Table 4: SAMPA vs. IPA Equivalences.

in the pronouncing dictionaries described in section 4, except for the vowel "/3/" that only occurs in diphthong. The phonetic repertoire of Faroese includes the following 12 diphthongs: **EA**, **OA**, **UJ**, **EJ**, **aJ**, **aW**, **OJ**, **OW**, **3W**, **EW**, **9W** and **9J**. Summing the 41 individual phonemes in Table 3, plus the 12 diphthong, plus seven phonemes with pre-aspiration (**Hb**, **Hd**, **HdZ**, **Hg**, **Hp**, **Ht**, **HtS**), we have a total of 60 phonemes. That is the list of 60 phonemes that are included in the dictionaries presented in section 4. To see an equivalence between our SAMPA symbols versus the IPA phonemes, please see Table 4.

## 5 Language Models

As it was mentioned in section 4, our "Faroese Language Models with Pronunciations" is a set of n-gram language models of distinct sizes that were created using the Faroese text provided in the BLARK, as it provides with text from newspaper articles, parliamentary speeches, books and more. The normalization process of that text included to change everything to lowercase, allow only characters belonging to the Faroese alphabet and removing punctuation marks.

The resulting text has a length of more than half million lines of text (106.3MB approximately). The text was used to create a 3-gram (recommended for decoding) and a 4-gram (recommended for re-scoring) language models with the SRILM toolkit (Stolcke, 2002). Both the 3-gram and 4-gram models come in pruned and unpruned versions. It also includes a 6-gram language model in binary format suitable for ASR experiments with the NeMo toolkit. In particular, this model was created using KenLM (Heafield, 2011). It is important to mention that all the words present in any of the language models are present in the pronouncing dictionaries for the east and central variants of Faroese (see section 4).

## 6 Results

Table 5 shows a comparison of the Word Error Rate (WER) obtained with the acoustic models presented in section 3 with the exception of the PocketSphinx models as discussed in section 3.4.

The NeMo results include the WER obtained using the 6-gram language model (LM) presented in section 5 as well as the WER obtained with no language model at all. The Kaldi results include the WER obtained with Hidden Markov Models (HMM) only and the WER obtained with the LSTM network. As it can be seen, the best results are obtained with the WAV2VEC2 model, which is not a surprise as it is well known that it can achieve acceptable results with less than 1 hour of speech data. What is remarkable indeed, is the gap of performance between WAV2VEC2 and the other systems.

In addition to this, based on our previous experience (Hernandez Mena et al., 2020; Mena et al., 2022), it is also remarkable that the WER obtained with NeMo using a language model and the WER obtained with Kaldi using the LSTM are so close to each other despite of the relatively low amount of training data. This fact reveals that the train-

| Corpus Portion | NeMo SP No LM | NeMo SP With LM | Kaldi HMM | Kaldi LSTM | WAV2VEC2 XLRS-53 |
|---|---|---|---|---|---|
| **Dev** | 20.51% | 13.66% | 20.60% | 12.22% | **5.56%** |
| **Test** | 22.81% | 15.95% | 23.44% | 14.04% | **7.60%** |

Table 5: WER Results.

ing method described by Huang et al. in 2020 and the use of speed perturbation for training are really effective in NeMo.

On the other hand, Table 6 shows the results obtained with the newest system Whisper (Radford et al., 2022). Whisper is a transformer-based speech recognition system trained with 680k hours of transcribed data in multiple languages. Whisper is also a multitask system able to perform multilingual speech recognition as well as speech translation and language identification. According to the original paper (Radford et al., 2022), the training set that Whisper uses for translation includes 46 hours of Faroese. Based on this, we decided to test Whisper in its distinct sizes with no fine-tuning step and using the development and test portions of the Ravnursson corpus. As it can be seen in Table 6, we obtained terribly bad WER results, revealing that Whisper needs to be fine-tuned prior to recognize Faroese data; unfortunately, this is beyond the scope of this paper but it will tackle as further work.

| Whisper Size | Dev WER | Test WER |
|---|---|---|
| **Tiny** | 113.4% | 116.7% |
| **Base** | 112.61% | 113.07% |
| **Small** | 128.05% | 132.64% |
| **Medium** | 116.34% | 119.3% |
| **Large** | 105.93% | 110.25% |

Table 6: Whisper WER Results.

## 7 Conclusions

A major development of Faroese ASR is presented in this work. The Ravnursson project has produced a corpus of 109 hours of transcribed speech and acoustic models for WAV2VEC2, NeMo, Kaldi and PocketSphinx have been developed. Furthermore, the project has also produced a set of n-gram language models of distinct sizes and pronunciation dictionaries in Faroese suitable for ASR experimentation. Quality assessment of the

acoustic models are shown in Table 5 where the best results of 7.60% WER was achieved by the WAV2VEC2 model. Another interesting result is shown in Table 6 demonstrating that a fine-tuning step is needed for Faroese for the multi-lingual ASR system Whisper.

Faroese ASR is no longer under-developed due to this work. The project has lowered the technological threshold for implementing ASR solutions for Faroese in industry and for studying the Faroese language using ASR as a tool. With all the results made available with open licenses, there is no good reason why Faroese ASR should not be included in standard language technology software in the future.

## 8 Further Work

As further work, it is clear to us that we have to explore acoustic models with the new parameter versions of WAV2VEC2 such as 300m, 1B and 2B; as well as the Whisper system with a fine-tuning step in Faroese in order to keep improving our WER results. Another future challenge is to add more Faroese data to our models, including conversational speech.

### Acknowledgments

This project was made possible under the umbrella of the Language Technology Programme for Icelandic 2019-2023. The Programme, which is managed and coordinated by Almannarómur, is funded by the Icelandic Ministry of Education, Science and Culture.

Special thanks to the Ravnur Project for making their "Basic Language Resource Kit" (BLARK 1.0) publicly available.

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
