# OpenReview forum: "ASR Language Resources for Faroese"
_NoDaLiDa/2023/Conference — NoDaLiDa 2023_

### Official Review · Reviewer_1AZ8 · 2023-03-08
**Basic support for Faroese ASR**

**Rating:** 7
**Confidence:** 5

**Review:**

This paper presents a number of language resources for Faroese. The resources are a corpus extracted from a BLARK for Faroese, phonetic dictionaries, LMs and AMs. Typically, resources would refer to corpora or other data collections, but with the advent of transfer learning, like the authors do with QuartzNet and wav2vec2 models, models themselves should also be considered resources.

In general, there is too much introduction in this paper and too little information on the design choices. The design choices explained in the paper are good choices, but I am missing information about how the 10sec segmentation was done,  why only 106h rather than the full 135 hours were not used to create the corpus, why QuartzNet instead of Citrinet, Conformer or other NeMo models.

If you did speed perturbation on the training data for QuartzNet, why not for the Kaldi models and did you do data augmentation for wav2vec2?

Finally, you skip a discussion and go right from results to conclusion. with no error analysis or explanation why whisper performs so poorly on your Faroese data. The paper would benefit greatly from a discussion on whether or not dictionaries and n-gram LMs are of much relevance as language resources for ASR, since the lexicon-based ASR models perform quite poorly by comparison to wav2vec2? And now that you have the basic resources for Faroese ASR in place, what is the next step to take Faroese from fragmentary/weakly supported to moderately or well-supported? What are the limits of the resources you have made available? You would do well to acknowledge the lack of conversational and spontaneous speech and the acknowledgement of the limiting factors reveals to you what the next steps are.

In summary, you have done good and valuable work for a small language in sore need of attention, but your documentation is lacking a bit and the scientific value is much less since there is no discussion with any insights or future work.

NB! Your anonymisation fails in line 700 and while the URL for the Kaldi implementation does not work, I can identify you by googling 'faroese kaldi' and locating the github page.

**Paper Type:**

Long paper

---

### Official Review · Reviewer_MLfW · 2023-03-08
**Excellent work**

**Rating:** 8
**Confidence:** 4

**Review:**

This is a very good paper that describes the latest resources for Faroese ASR. It is well-written, clear, and definitely helpful for anybody who would like to work with Faroese ASR. I fully agree with the claim that authors make in conclusions - Faroese ASR is no longer under-developed due to this work. (line 746)
Still I have a few suggestions.
1. Authors write that PocketSphinx results are not included as it is no longer competitive. (line 686) I would like to see these results even if they are worse than Whisper's results. Why are you dealing with PocketSphinx at all, if you are not reporting results?
2. It is obvious that it is almost impossible to fully anonymize such a paper. It refers to a particular project etc. It is fine for me, I think authors have done almost all that is necessary. But I have a practical suggestion - next time when you submit a paper for a blind review, please do not use phrases that explicitly reveal you. For example - According to our previous experience (Hernandez Mena et al., 2020). You can refer to your previous publications like this - (Authors, 2020)
3. refer to CC licenses in several places. I think that you should write "CC BY 4.0", which is the official abbreviation, instead of "CCBY4". (lines 396, 427, 482)
4. when you say that models are available, please also add a link in the footnotes, it will help readers to models.
5. there might be a spelling error in line 672, shouldn't it be "it also includes" instead of "it is also included"
6. there is a technical glitch in line 666, shouldn't it be "106.3 MB" instead of "106.3M B". Check spaces and italic.
7. Numbers should be aligned to the right not reft. See tables 1, 5 and 6. Even better would be to align to the right by the decimal point.



**Paper Type:**

Long paper

---

### Official Review · Reviewer_K7ia · 2023-03-10
**Review of the paper "ASR Language Resources for Faroese"**

**Rating:** 7
**Confidence:** 4

**Review:**

The paper introduces a set of new language resources in Faroese and their implementation in ASR development for Faroese.
The topic is certainly relevant, as the development of ASR technology for the Faroese language is crucial to enable speakers to interact with the technology in their native language and will contribute to the survival of the language in the information society.
The BLARK methodology adopted for Faroese is a welcomed approach to building necessary resources and infrastructure for language technology needs.
The corpus introduced in the paper is a good starting point for ASR development. Training different acoustic models in the development process seems justified, however, in the comparison of different acoustic models, it was not surprising that WAV2VEC2 models outperformed the older methods.

Remarks:
- The title of chapter 3.1 Basic Language Resource Kit (BLARK) for Faroese is confusing as it leads a reader to expect an overview of the Faroese language resources, however, it mostly introduces the collection of the speech corpus for ASR.
- Representing consonants and vowels in a single table is not correct as the manner and place of articulation do not apply to the classification of vowels.
- line 707: Huang et al. - year of publication missing

**Paper Type:**

Long paper

---

### Decision · Program_Chairs · 2023-03-17

Accept